# Route to high-energy density polymeric nitrogen *t*-N via He—N compounds

Yinwei Li[1], Xiaolei Feng[2,3], Hanyu Liu[4], Jian Hao[1], Simon A.T. Redfern [3,5], Weiwei Lei[6], Dan Liu[6] & Yanming Ma[2,7]

Polymeric nitrogen, stabilized by compressing pure molecular nitrogen, has yet to be recovered to ambient conditions, precluding its application as a high-energy density material. Here we suggest a route for synthesis of a tetragonal polymeric nitrogen, denoted *t*-N, via He-N compounds at high pressures. Using first-principles calculations with structure searching, we predict a class of nitrides with stoichiometry $HeN_4$ that are energetically stable (relative to a mixture of solid He and $N_2$) above 8.5 GPa. At high pressure, $HeN_4$ comprises a polymeric channel-like nitrogen framework filled with linearly arranged helium atoms. The nitrogen framework persists to ambient pressure on decompression after removal of helium, forming pure polymeric nitrogen, *t*-N. *t*-N is dynamically and mechanically stable at ambient pressure with an estimated energy density of ~11.31 kJ/g, marking it out as a remarkable high-energy density material. This expands the known polymeric forms of nitrogen and indicates a route to its synthesis.

[1] School of Physics and Electronic Engineering, Jiangsu Normal University, Xuzhou 221116, China. [2] State Key Laboratory of Superhard Materials, College of Physics, Jilin University, Changchun 130012, China. [3] Department of Earth Sciences, University of Cambridge, Downing Street, Cambridge CB2 3EQ, UK. [4] Geophysical Laboratory, Carnegie Institution of Washington, Washington, DC 20015, USA. [5] Center for High Pressure Science and Technology Advanced Research (HPSTAR), Shanghai 201203, China. [6] Institute for Frontier Materials, Deakin University, Waurn Ponds, VIC 3216, Australia. [7] International Center of Future Science, Jilin University, Changchun 130012, China. Correspondence and requests for materials should be addressed to Y.L. (email: yinwei_li@jsnu.edu.cn) or to H.L. (email: haliu@carnegiescience.edu) or to S.A.T.R. (email: satr@cam.ac.uk)

Nitrogen-rich compounds have attracted considerable attention due to their potential as high-energy density materials (HEDM), with considerable energy storage or release upon rearrangement of bonding within and between associated nitrogen molecules. The triply-bonded di-nitrogen molecule, $N\equiv N$, displays one of the strongest chemical bonds known in nature. Thus, diatomic nitrogen, as encountered at ambient conditions, is chemically inert and a huge amount of energy is needed to transform its triple bond into singly bonded, N–N, nitrogen. Conversely, the transformation of compounds containing singly bonded nitrogen into triply bonded material is accompanied by a huge release of energy. While crystalline nitrogen at cryogenic temperatures and ambient pressure is dominated by weak van der Waals forces that are responsible for packing of its molecular structure, polymeric nitrogen-rich crystals result when $N_2$ is transformed into single- or double-bonded poly-nitrogen crystals, which are metastable with respect to the triple bond molecule. The large energy difference between a molecule of $N_2$ bonded with the single bond (~160 kJ/mol) and the triple bond (approaching 1 MJ/mol) means that this transformation has an energy release higher than that of any other known non-nuclear explosive reaction.

The synthesis of singly bonded polymeric nitrogen compounds is particularly challenging, however. While azides, with linear $N_3$ radicals, have been synthesized rather easily for well over a century, true poly-nitrogen molecules or clusters are highly metastable and can, thus far, only be produced by transformation of molecular nitrogen at pressures >110 GPa. A singly bonded phase of nitrogen was first predicted from computational simulation a quarter century ago[1], but it took more than a decade before this cubic gauche allotrope of nitrogen (cg-N) was synthesized by high-pressure/temperature methods at >110 GPa and found to be metastable to lower pressure at least down to 25 GPa[2,3]. Several new polymeric nitrogen forms have subsequently been predicted at high pressures, such as the Cmcm chain[4] phase, chaired web[5], cis–trans chain[6], layered Pba2[7], helical tunnel $P2_12_12_1$[7,8], cage-like diamondoid $N_{10}$[9] and P4/nbm[10]. Of these, the existence of the layered Pba2 phase, predicted computationally, was subsequently confirmed by experiment using laser-heated diamond anvil cells at pressures between 120 and 180 GPa[11]. Other allotropes of nitrogen formed of $N_6$ or $N_8$[12–14] molecules containing single and double bonds have also been suggested by more recent computational studies. While these materials are also only stable at high pressure, metadynamics calculations suggest that they may be thermally metastable down to ambient conditions. This has yet to be confirmed experimentally, however, and the challenge remains to recover and preserve polymeric nitrogen compounds at room temperature and pressure.

Nitrogen-rich compounds provide an alternative route to the synthesis of materials with single or double nitrogen bonds at ambient conditions. Attention has focused, thus far, on the lower pressure limits of thermodynamic stability of compounds such as $LiN_3$[15,16], $NaN_3$[17], $KN_3$[18–21], $CO–N_2$[22], $C_3N_{12}$[23], $LiN_5$[24], and $N_2H$[25]. Boron nitride-related materials have also been explored as potential high-energy density materials[26]. The high-pressure structures predicted by such approaches do not, however, comprise pure polymeric nitrogen molecules, but are instead boron–nitrogen intercalated compounds. An alternative approach, adopted here, is to search for nitrogen-rich compounds which comprise polymeric nitrogen molecules co-existing with other elements, but in which the secondary atoms do not form a direct part of the polymer network. Such secondary atoms may act as a template or former but could be removed in experimental procedures after a first stage of synthesis, in which they are present, to produce a pure nitrogen polymolecular material.

Ab initio structural prediction can now reliably search for previously unknown or unsynthesized crystalline materials. The work on boron nitrides[26] provides a useful illustration of the approach. A similar approach can be used to explore the phase stability and structure-property relations of compounds in the $He_xN_y$ system. The possibility of stable compounds of helium, in this case with nitrogen, also impinges upon another long-standing question—the extent to which helium is noble and unable to form compounds due to its intrinsic inertness. Recently, Dong et al.[27] demonstrated the existence of He-Na-electrides in their discovery of a stable $Na_2He$ compound—a fluorite-like structure that is stabilized by high pressure. The possibility that helium can form more complicated compounds has also been indicated by the suggestion that it can react with $H_2O$ to form bonded high-pressure solids with structural helium, rather than simply filling interstices in an ice[28]. Similar indications from noble gas nitrides, stabilized at high pressure, have also been noted in earlier work[29,30]. For helium–nitrogen compounds in particular, the closed-shell van der Waals compound $He(N_2)_{11}$ was crystallized at high pressure from a wide composition range of $N_2$–He mixtures, but not found to contain polymeric nitrogen[31].

Here, we report ab initio structure prediction calculations on the entire composition range of the helium–nitrogen system reveal the existence of a number of energetically stable $He_xN_y$ compounds as a function of pressure. In particular, phonon calculations demonstrate three polymorphs of $HeN_4$ that are dynamically stable over a wide range of pressures. They show increased polymerization and single-bond character on increasing pressure. The highest-pressure polymorph comprises a completely polymerized nitrogen framework structure with interstitial helium. We show that the nitrogen framework can be preserved to ambient pressure after removal of helium, to form a new pure nitrogen polymeric solid. This nitrogen polymeric solid, denoted t-N, is dynamically stable at ambient pressure.

## Results

**Ab initio structure prediction.** Ab initio structure predictions for $He_xN_y$ were performed using the particle swarm optimization technique implemented in the CALYPSO code[32,33]. CALYPSO has been used to investigate a great variety of materials at high pressures[34–40]. We have performed structure searches on $He_xN_y$ ($x = 1-4$ and $y = 1-9$) at 0, 25, 50 and 100 GPa with maximum simulation cells up to 4 formula units (f.u.) for each fixed-composition. Figure 1a shows the results of structure search at ambient pressure, in terms of the enthalpies of formation of $He_xN_y$ compounds, both with and without the inclusion of vdW corrections in the calculation of enthalpy. Here the formation enthalpy is defined as $\Delta H = H(He_xN_y) - [xH(He) + yH(N)]$ in unit of eV/atom. The enthalpy across the composition range is rather flat, with no obvious compound more stable than the others and all showing a positive enthalpy of mixing compared with a simple chemical mixture of helium and nitrogen. On increasing pressure, however, the stabilization of a nitrogen-rich compound of stoichiometry $HeN_4$ becomes apparent, as can be seen for the equivalent figure for calculations performed at 25 GPa (Fig. 1b) and 50 GPa (Fig. 1c). $HeN_4$ is enthalpically stable with respect to break down to He + 4 N under these conditions. The inset in Fig. 1d indicates that $HeN_4$ is stable with respect to a chemical mixture of He and $N_2$ at pressures as low as 8.5 GPa. It is interesting to note that $HeN_4$ becomes enthalpically unstable with respect to breakdown to He + 4 N above 51 GPa due to the calculated phase transition of pure molecular $N_2$ to cg-N at about 54 GPa, with an associated sharp decrease of its enthalpy (Supplementary Figure 1). In fact, experiments have shown that the

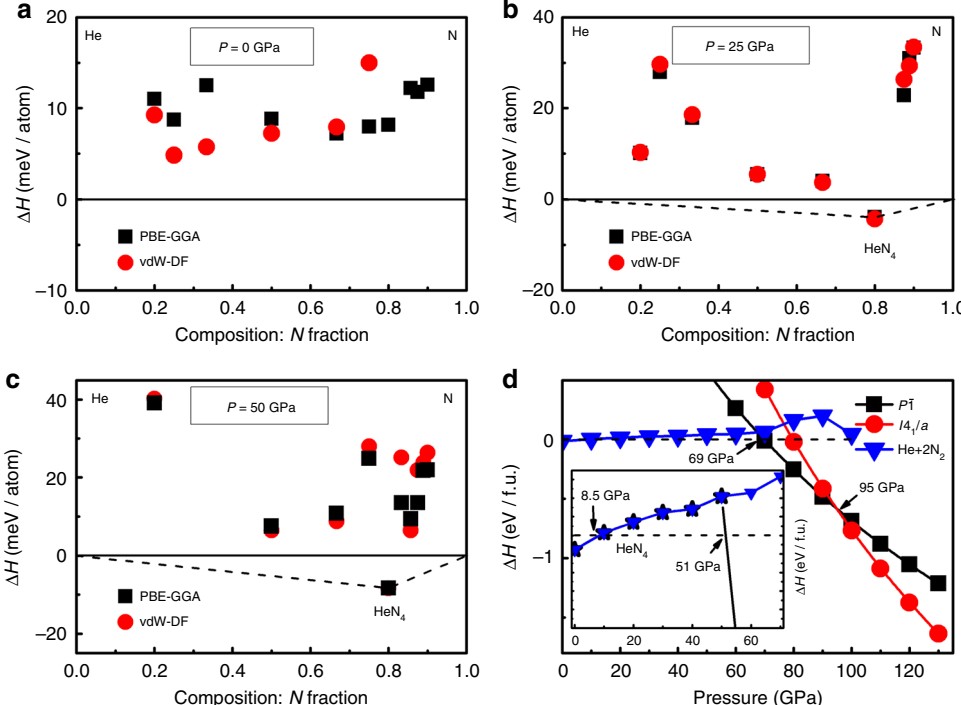

**Fig. 1** Thermodynamic stability of the predicted structures. Results from structure searching at 0 (**a**), 25 (**b**) and 50 (**c**) GPa. Convex hulls are shown as continuous lines, with (red) and without (black) the inclusion of vdW corrections. $hcp$-He[50] at 0, 25 and 50 GPa, $\alpha$-$N_2$[51] at 0 GPa and $\varepsilon$-$N_2$[52,53] at 25 and 50 GPa were adopted in the calculation. **d** Calculated enthalpy curves of the $I4_1/a$ and $P\bar{1}$ structures with respect to the $C2/c$ structure for $HeN_4$ as a function of pressure. The decomposition enthalpy (blue triangles) into $hcp$-He plus molecular $N_2$ ($\alpha$-$N_2$ at 0 GPa and $\varepsilon$-$N_2$ above 0 GPa) was also plotted. Inset in **d** is an enlarged view of the decomposition enthalpy in pressure range 0–70 GPa. Black stars in inset represents the decomposition enthalpy curves of $HeN_4$ into (He + 4 N) as a function of pressure, where $cg$-N was considered at pressures above 54 GPa

existence of a large activation barrier prevents the transformation from molecular $N_2$ to $cg$-N, which means that molecular $N_2$ can exist metastably up to 110 GPa[2,3]. Taking the formation enthalpy calculations into account, the stable pressure range of $HeN_4$ extends up to at least 110 GPa (Fig. 1d).

**Polymorphism of $HeN_4$ at high pressure**. Across the pressure range, three stable structures of $HeN_4$ are found, corresponding to $C2/c$, $P\bar{1}$ and $I4_1/a$ symmetries on increasing pressure. The optimized lattice parameters of these three structures at selected pressures have been listed in Supplementary Table 1. The lowest-pressure, monoclinic, phase comprises two interpenetrating lattices of He atoms and $N_2$ molecules, with the molecules aligned in a canted arrangement sub-parallel to the $a$-axis of the monoclinic unit cell, and a N - N bond length of 1.107 Å at 25 GPa, corresponding to a triple bond of nitrogen. There is no evidence of polymerization of the nitrogen dimer network (Fig. 2a), nor of charge transfer between the nitrogen molecules and the helium atoms (Supplementary Table 3). The shortest inter-molecular nitrogen–nitrogen distance is 2.63 Å. The phonon dispersion curves for this structure confirm the molecular nature of $N_2$, resulting in a very high energy flat optic mode lying above 70 THz associated with the intramolecular nitrogen stretch (Fig. 3a). All other modes lie at low frequency, due to the weak $N_2$–$N_2$, $N_2$–He and He–He interactions. These results are in consistent with our calculated interaction energy between He and N atoms. As shown in Supplementary Table 6, the low values indicate the rather weak interaction between He and N atoms and $HeN_4$ is typical a vdW closed shell compound. Electronic band structures show that the $C2/c$ phase of $HeN_4$ is an insulator with a large indirect band gap of 5.33 eV at 25 GPa (Supplementary Figure 2a). The existence of

this phase does, however, extend the known range of expected helium compounds, and bears further investigation in its own right within the context of understanding the limits of nobility of light noble gas elements. Given it does not represent a likely high energy density nitride, however, we have not explored its likely behavior on removal of the helium from the structure, as we do below for the tetragonal compound.

Increasing pressure leads to the transformation to a triclinic $P\bar{1}$ polymorph at 69 GPa in which the He atoms are arranged in layers parallel to (001). Within these layers the He atoms run in zig-zag chains parallel to the $a$-axis (Supplementary Figure 3). The nitrogen atoms form a homocyclic double pentagonal ring $N_8$ molecule with nitrogen atoms lying on four crystallographically distinct sites which we label N1, N2, N3, and N4 with all N–N distances lie around 1.3 Å. The $N_8$ molecules are planar and all aligned parallel to each other canted by a small angle away from the $(1\bar{1}0)$ plane. Mulliken population calculations indicate no charge transfer between nitrogen atoms and helium (Supplementary Table 3), once again demonstrating that the helium and nitrogen networks are rather independent of one another. Charge transfer is seen, however, from N1 to N2, N3 and N4 within the cyclic N8 molecule, indicative of the single N1–N1 bond and higher strength N1–N2, N1–N3, N2–N4 and N3–N4 bonds. Such charge transfer also results in the absence of lone pair of two N1 atoms, as shown in Fig. 2e. The $P\bar{1}$ phase was found to have a small indirect band gap of 1.77 eV at 80 GPa (Supplementary Figure 2b).

The highest-pressure polymorph, tetragonal $I4_1/a$, becomes energetically most stable at 95 GPa. In this structure all nitrogen atoms are symmetrically equivalent and each is bonded to three other nitrogen atoms in a polyatomic network structure. Rings of

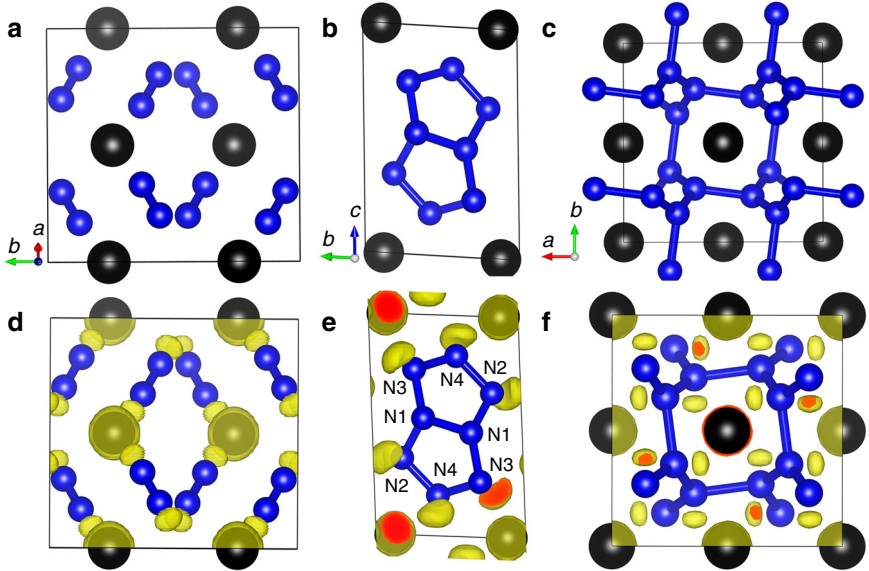

**Fig. 2** Crystal structures of HeN$_4$. Energetically favorable structures of HeN$_4$ with space groups $C2/c$ (**a**), $P\bar{1}$ (**b**) and $I4_1/a$ (**c**). Small blue and large black spheres represent N and He atoms, respectively. Three-dimensional valence electron localization functions with isosurface value of 0.95 for the $C2/c$ structure (**d**), and of 0.9 for the $P\bar{1}$ (**e**) and the $I4_1/a$ (**f**) structures

ten nitrogen atoms are linked by 8-fold rings in a three-dimensional framework that displays channels running parallel to each crystallographic axis (Fig. 2c and Supplementary Figure 4). Along $a$ and $b$-axes, these channels are formed by the 10-fold rings, while along $c$-axis they are formed of 8-fold rings. Each nitrogen is bonded to three others with two at a distance of 1.28 Å with the third lying slightly further away at 1.43 Å. The three bond angles made at each central nitrogen atom to the adjacent triplet of nitrogen atoms are 110°, 114° and 132°. No charge transfer between nitrogen and helium was found (Supplementary Table 3), indicating weak interaction between the nitrogen single bond network structure and the helium atoms lying within the channels in the structure. The phonon dispersion curves for the $I4_1/a$ structure indicate rather a broad phonon density of states and strong dispersion of both optic and acoustic phonons, with some indication of incipient softening at the Z-point of the Brillouin zone, indicative of the potential for a displacive phase transition to lower symmetry doubled unit cell structure. The band structure calculations reveal an indirect band gap of 3.4 eV at 100 GPa, indicating the semiconducting nature of the $I4_1/a$ phase (Supplementary Figure 2c).

**Tetragonal polymeric nitrogen $t$-N.** Given the weakness of the interaction between the channel-filling helium atoms and the nitrogen polymeric framework of the $I4_1/a$ HeN$_4$ structure, we explored the effect of removing He atoms from this tetragonal phase, and the resultant nature of the bare nitrogen structure, which we denote $t$-N (tetragonal nitrogen), at ambient pressure. This simulates the new potential chemical route to synthesis of an ambient pressure polymeric nitrogen structure. Metastable preservation of the $I4_1/a$ HeN$_4$ structure to lower pressure, followed by extraction of helium from the channel sites of that structure, would result in the new ambient-pressure $t$-N phase. In fact, a similar methodology has recently been used to synthesize a new allotrope of silicon[41]. Experiments demonstrated that Na atoms can be removed from the high-pressure compound Na$_4$Si$_{24}$ by a thermal degassing or volatilization process, forming a new orthorhombic allotrope Si$_{24}$[41]. Interestingly, Na$_4$Si$_{24}$ consists of a channel-like $sp^3$ silicon structure filled with linear Na chains, and

hence is structurally similar to $I4_1/a$ HeN$_4$. Therefore, we conclude that it is likely possible to synthesize $t$-N by removing He atoms from $I4_1/a$ HeN$_4$ via diffusion and degassing along the channels.

**Discussion**

The $I4_1/a$ pure nitrogen structure retains the same nitrogen topology and framework as that of the $I4_1/a$ HeN$_4$ phase, but with helium atoms (obviously) absent. The volume collapse is found to be only 6.6% by removing He atoms, which provides further support for the weak interactions between He atoms and nitrogen polymeric framework. The framework is identical, but depressurization to ambient pressure results in increases of bond lengths in the low-pressure lower-density structure. In the $t$-N $I4_1/a$ structure, nitrogen atoms remain symmetrically equivalent, on the Wyckoff 16$f$ position, and now each nitrogen is bonded to three others with two at a distance of 1.37 Å with the third lying slightly further away at 1.61 Å. $t$-N was calculated to have an indirect band gap of 1.7 eV.

The calculated phonon dispersion curves (Fig. 4b) and elastic constants (Supplementary Table 4) confirm the dynamical and mechanical stabilities of $t$-N at ambient pressure. This fact demonstrates that the removal of He atoms from HeN$_4$ does not affect the lattice stability of the N framework. Our first-principles molecular dynamics simulations show that the $I4_1/a$ structure is thermally stable up to 1000 K (Supplementary Figure 5). The existence of such thermal stability, together with the high-pressure range over which we predict $t$-N to be stable, indicate that it might well be possible to synthesize this compound within the diamond anvil cell.

We have carried out a topological analysis of all-electron charge density of $t$-N using the quantum theory of atoms-in-molecules. Note that one of the N-N bond-lengths (1.61 Å) in the $t$-N $I4_1/a$ structure, is longer than the value of 1.41 Å found for the $cg$-N structure at ambient pressure. To investigate these bonding feature, we have calculated the Laplacian $\nabla^2\rho(r)$ of N-N bonds in the $t$-N $I4_1/a$ structure, as shown in Table 1. The $\nabla^2\rho(r)$ of the short and long nitrogen interactions are $-26.306$ and $-0.869\,e^-\,\text{Å}^{-5}$, respectively. This shows the obvious hints of

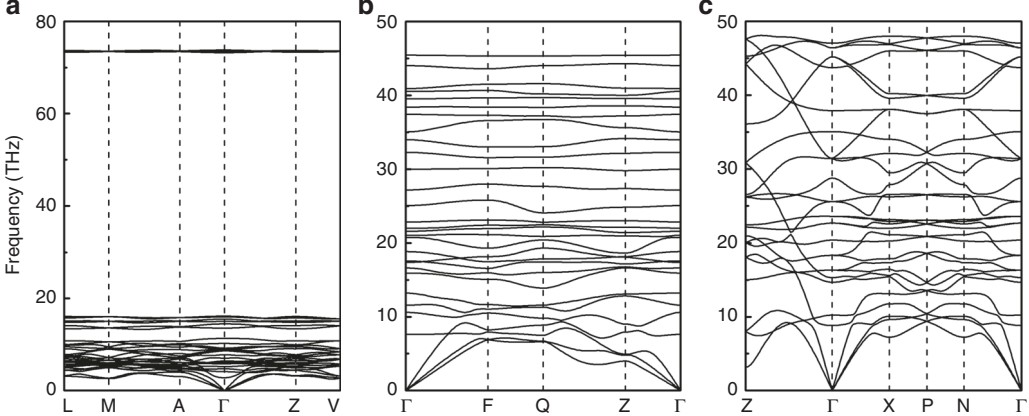

**Fig. 3** Phonon dispersions of HeN$_4$. Phonon dispersions of HeN$_4$ with the C2/c structure at 25 GPa (**a**), the P$\bar{1}$ structure at 80 GPa (**b**) and the I4$_1$/a structure at 100 GPa (**c**)

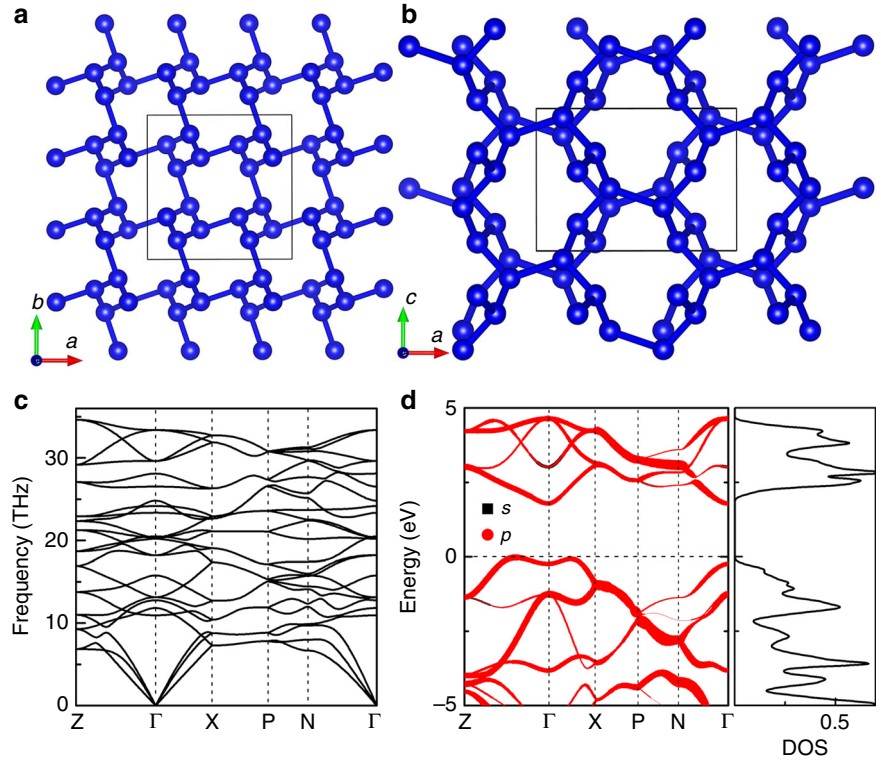

**Fig. 4** Crystal structure, phonon dispersion and electronic band structure of t-N. **a** Crystal structure of t-N viewed along c-axis and **b** b-axis. **c** and **d** are phonon dispersions and electronic band structure of t-N at ambient pressure, respectively

covalent interactions between nitrogen atoms. These results indicate the t-N I4$_1$/a phase is indeed a 3D nitrogen polymeric bonded structure.

Our results extend the boundaries of known helium-compounds, in particular suggesting that helium may form compounds with elements such as nitrogen at sufficiently high pressures. Given the generation of abundant helium in stars like the sun, leading to the fact that it is the second most abundant element in the Milky Way galaxy, for example, this suggests that helium compounds may indeed exist deep within icy moons and planets, as phases previously unrecognized. The existence of helium-nitrogen phases, with nitrogen the seventh most abundant element in our galaxy, seems increasingly likely in the light of our calculations.

**Table 1 Topological analysis of N–N bonding in the I4$_1$/a structure at ambient pressure**

|       | $d$(Å) | Multiplicity | $\rho$ (e$^-$Å$^{-5}$) | $\nabla^2\rho$ (e$^-$Å$^{-5}$) |
|-------|--------|--------------|------------------------|--------------------------------|
| N–N   | 1.37   | 2            | 2.42                   | −26.306                        |
|       | 1.61   | 1            | 1.41                   | −0.869                         |

Unlike the ionic nature of He–Na bonds predicted in Na$_2$He[27], the bond between helium and nitrogen remains weak in the materials under study here, which gives rise to the possibility to generate polymeric pure nitrogen by extraction of the

helium within the structure, leading to the new *t*-N polymeric nitrogen phase. Total energy calculations using DFT show that *t*-N possesses an enthalpy higher than *cg*-N by 0.14 eV/atom at ambient pressure, thus the reversion of *t*-N to gaseous $N_2$ would be accompanied by very significant release of energy, leading to the suggestion that this structure could represent a very high energy density solid, unlike any previously seen. At the PBE-GGA level, the energy difference of 1.642 eV/atom between *t*-N and diatomic nitrogen corresponds to an energy density of ~11.31 kJ/g, significantly higher than that (9.7 kJ/g) predicted for *cg*-N.

Finally, we considered the relationship of *t*-N to the experimentally reported $He(N_2)_{11}$ phase[31], for which a later X-ray diffraction study[42] proposed a hexagonal structure. This has 46 atoms in the unit cell with orientationally disordered $N_2$ molecules. We have constructed a model ordered structure for this phase of $He(N_2)_{11}$ and compute a positive formation enthalpy of ~ 0.02 eV/atom as compared to a mixture of He and $N_2$ at 25 GPa. The large number of atoms in the cell and the orientational disorder of $N_2$ prevents us from arriving at a better measure of its stability with respect to our predicted $HeN_4$ phase within the present ab initio framework. Nevertheless, we anticipate that $HeN_4$ is favored at high pressures in He-rich mixtures and may be synthesizable at high pressures. Moreover, we believe our results will stimulate further experiment and theory to revisit this important system at high pressures.

In summary, we have performed extensive first-principles simulations on the crystal structures and their stabilities of He-N system at high pressures. These computational simulations indicate the thermodynamic stability of a previously unrecognized class of nitrogen-rich nitrides with a stoichiometry of $HeN_4$, stable relative to an equivalent mixture of solid He and $N_2$ above 8.5 GPa. The $HeN_4$ compound adopts a structure comprising a polymeric channel-like $sp^3$ nitrogen framework filled with chains of helium atoms at pressures above 95 GPa. Further calculations indicated that the nitrogen framework can be preserved to ambient pressure on decompression after the removal of helium atoms, to form a pure polymeric form of nitrogen denoted as *t*-N. Given the indicated metastable persistence of *t*-N at ambient pressure it appears that $I4_1/a$ $HeN_4$ may be regarded as a clathrate-type structure. Since $I4_1/a$ $HeN_4$ is more stable than a mixture of He + $N_2$ at high pressure, it appears that dense packing plays a critical role in helping to stabilize this structure. In these respects, it is akin to the previously reported stable $H_2O$–He clathrate structure[28]. Topological analysis indicates the *t*-N $I4_1/a$ phase is a 3-D nitrogen polymeric bonded structure. This *t*-N is dynamically and mechanically stable at ambient pressure and has an estimated energy density of ~11.31 kJ/g, thus represents a very high-energy density material. Most importantly, these results indicate a route to the synthesis of high energy density polymeric nitrogen, via the He-bearing phase, since the He-N bonding interaction is weak and the channel structure of $HeN_4$ and *t*-N could facilitate the expulsion of He in a final synthesis step to generate polymeric nitrogen. Such synthesis has yet to be demonstrated experimentally, but our molecular dynamics simulations indicate that *t*-N is dynamically stable to high temperature, allowing the possibility of high-temperature methods in any processing route.

## Methods

**Ab initio calculations**. The fixed-composition structure search was considered converge when >1000 successive structures were generated after a lowest energy structure was found. Ab initio structure relaxations were performed using density functional theory (DFT) with the Perdew–Burke–Ernzerhof (PBE) generalized gradient approximation (GGA) implemented in the Vienna ab initio simulation

package[43]. Van der Waals density functional[44–46] method was used in the optimization of selected structures. The all-electron projector augmented wave[47] pseudopotentials with $1s^2$ and $2s^2 2p^3$ valence configurations were chosen for He and N atoms, respectively. An energy cut-off of 500 eV and a Monkhorst-Pack Brillouin zone sampling grid with a resolution of 0.3 Å$^{-1}$ were used in the structure searches. The low enthalpy structures found were then re-optimized with denser grids better than 0.2 Å$^{-1}$ and a higher energy cutoff of 700 eV. We performed phonon calculations by the direct supercell method as implemented in the PHONOPY[48] program to determine the dynamical stability of the studied structures. The crystal structures and electron localization functions (ELF) were drawn using the VESTA[49] software.

**Molecular dynamics**. We also performed first-principles molecular dynamics (MD) simulations to examine the thermal stability of the *t*-N $I4_1/a$ structure using the canonical *NVT* (*N*-number of particles, *V*-Volume and *T*-temperature) ensemble. A $2 \times 2 \times 2$ supercell of 128 atoms was employed. The MD calculations were performed at temperatures of 300 and 1000 K. Each simulation consists of 10,000 time steps with a time step of 1 fs.

**Electron charge density**. We have carried out a topological analysis of all-electron charge density of *t*-N using the quantum theory of atoms-in-molecules. In this theory, a solid is defined by a zero-flux surface of the electron density gradient $\nabla \rho$ (*r*). The charge density distribution $\rho(r)$ and its principal curvatures (three eigenvalues of the Hessian matrix) located at the bond critical point (BCP), reveal information about the type and properties of bonding. The second derivative of the electron density (Laplacian) $\nabla^2 \rho(r)$ reveals information about the concentration of the electrons at the BCP. It has been shown that the $\nabla \rho(r)$ and $\nabla^2 \rho(r)$ at the BCP are directly related to the bond order and the bond strength.

**Data availability**. The structural (CIF) data for each predicted phase, and the output of an NVT-MD simulation of *t*-N at temperature ≈ 1000 K and pressure ≈ 1 atm, that support the findings of this study, are available in figshare with identifier DOI:10.6084/m9.figshare.5709826 (also available as https://figshare.com/s/823fcb2bd8ab014b3f07).

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

## Acknowledgements

Y.L. and J.H. acknowledge funding from the National Natural Science Foundation of China under Grant Nos. 11722433 and 11404148. X.F. and Y.M. acknowledge funding from Science Challenge Project at No. TZ2016001, the National Natural Science Foundation of China under Grant No. 11534003, and the National Key Research and Development Program of China under Grant No. 2016YFB0201200. Work at Carnegie was partially supported by EFree, an Energy Frontier Research Center funded by the DOE, Office of Science, Basic Energy Sciences under Award No. DE-SC-0001057. S.A.T. R. acknowledges the support of the UK Natural Environment Research Council under grant NE/P012167/1. The infrastructure and facilities used at Carnegie were supported by NNSA Grant No. DE-NA-0002006, CDAC. W.L. and D.L. acknowledge funding from the Australian Research Council Discovery Early Career Researcher Award scheme (DE150101617 and DE140100716). All the calculations were performed using the High Performance Computing Center of School of Physics and Electronic Engineering of Jiangsu Normal University.

## Author contributions

Y.L. designed the project and predicted new structures; Y.L., H.L. and S.A.T.R. conceived the research; Y.L., X.F. and H.L. performed the simulations; all authors analyzed and interpreted the data; Y.L., X.F., H.L. and S.A.T.R. wrote the paper.

## Additional information

**Competing interests:** The authors declare no competing financial interests.

