## [Peer Review File · Nature Communications]

Reviewers' comments:

Reviewer #1 (Remarks to the Author):

In this manuscript, the authors predict three novel helium-nitrogen compounds at high pressure through computational structure searching and first principles calculations. Furthermore, the highest pressure polymorph is examined in the context of removing the helium atoms, which leads to a new polymeric form of nitrogen, dubbed t-N, which is predicted to be recoverable to ambient pressure, and, through molecular dynamics simulations, predicted to be thermally stable up to 1000 K.

On the whole, this manuscript presents valuable new insights into helium compounds, and helium-nitrogen compounds in particular, as well as revealing a potential new single-bonded nitrogen structure that may be synthesizable in laboratory conditions. The analysis of bond lengths, charge density, electronic population and band structures, and phonon dispersion appears to be rigorous and sound, verifying the properties and stability of the predicted structures. There are, however, a few minor questions and comments which arise:

1. It is mentioned that the weak interactions between the helium and nitrogen atoms in the I41/a structure was a factor in considering whether the helium atoms could be removed and the nitrogen structure maintained. However, it also appears that the C2/c structure also has weak He-N interactions. It may be worth noting why the C2/c structure was not considered as well, whether it be due to the differences in the dimeric nitrogen in the C2/c structure versus the polymeric, ring structure in the I41/a structure, or some other reason.

2. There are a number of spelling errors which should be addressed, for example, “choosed” instead of “chosen”, “metstably” instead of “metastably”, “axes” instead of “axes”.

However, beyond these minor comments, I feel this paper presents enough interesting and novel content which falls within the scope of Nature Communications and the interests of its readers to recommend acceptance of this manuscript for publication.

Reviewer #2 (Remarks to the Author):

This paper predicts several new potential high-pressure phases of nitrogen using density functional theory and the particle swarm optimization algorithm. A couple features make this work particularly novel:

(a) the phases involve mixtures of N and He, particularly HeN₄. Such helium-containing compounds are rather new. Unlike some earlier examples, the helium does not appear to interact strongly with the nitrogens (in a covalent bonding sense), but it does appear important in stabilizing the structures.

(b) The highest pressure phase forms helium-filled channels, and molecular dynamics calculations predict it may still be stable even with the helium evacuated. If it were achievable experimentally, they predict this would be the highest energy-density form of nitrogen by a large margin.

This is an interesting contribution, the channel-containing nitrogen structure would be quite unusual, and while there is not experimental evidence for these phases yet, the results do suggest one might be able to make these compound by crystallizing a nitrogen helium mixture under pressure. So it may be publishable. However, there are some issues that need clarification before it would be suitable for publication.

1) The paper does not discuss the 1992 Nature report of an experimental He(N₂)₁₁ compound (DOI:10.1038/358046a0) or other nitrogen related inclusion compounds (DOI: 10.1063/1.3533957 & 10.1103/PhysRevB.64.184101). The He(N₂)₁₁ compound would be particularly interesting target for the authors to pursue, both in an attempt to clarify the structure of that experimentally reported structure and to substantiate their predictions for HeN₄.

2) The twin arguments that the helium is not strongly interacting with the nitrogen and that these phases would preferentially form over separate nitrogen and helium solids are perplexing. One would expect that a more densely packed nitrogen structure + pure He crystals would be more stable unless there is something special to the N-He bonding. So why does it form? Helium isn't exactly know for forming strong non-covalent interactions. It's hard to tell from the structure pictures--are any of these related to clathrate-type structures?

Several related comments & questions:

- I am not entirely clear what form of He/N are used in the formation enthalpies H(He) and H(N) when computing the stability. Are those crystalline? If so, which polymorph of N?

- Similarly, when the authors state that the low-pressure HeN₄ phase is stable all the way down to 8.5 GPa, is that stable relative to the epsilon molecular phase of nitrogen + crystalline He? It's not clear to me that the definition of stability used is one relative to the most stable phase at those same conditions. If it's not, then the important question is how much less stable is it than the thermodynamically preferred phase at those conditions? If it's too far above the stable molecular forms at those pressures, you won't have much hope of making even the metastable HeN₄ form.

- How strongly is the He interacting with nitrogen lattice in these different phases? One could easily quantify the interaction energy by comparing single-point enthalpy differences of the different lattices with and without helium in the structure, analogously to how they studied the I41/a phase without He.

3) Minor point: I have not done Mulliken population calculations in the solid state, but they seem a potentially poor judge of charge transfer given their strong reliance on the nature of the underlying orbitals. This is very clear in Gaussian basis sets, where for example the amount of Mulliken charge transfer in the water dimer varies dramatically with the basis set, including even the direction of the charge transfer:

Net Hartree-Fock charge transfer from hydrogen bond donor water to acceptor water molecule:

cc-pVDZ -0.0430

cc-pVTZ -0.0292

cc-pVQZ -0.0245

aug-cc-pVDZ +0.0279

aug-cc-pVTZ +0.0038

aug-cc-pVQZ -0.0383

The solid state Mulliken analog requires localizing the planewave density onto local atomic like orbitals, and I am concerned that the results would depend very strongly on how one localizes the orbitals.

Response to reviewer #1

REFeree#1 OVERVIEW. *In this manuscript, the authors predict three novel helium-nitrogen compounds at high pressure through computational structure searching and first principles calculations. Furthermore, the highest pressure polymorph is examined in the context of removing the helium atoms, which leads to a new polymeric form of nitrogen, dubbed t-N, which is predicted to be recoverable to ambient pressure, and, through molecular dynamics simulations, predicted to be thermally stable up to 1000 K. On the whole, this manuscript presents valuable new insights into helium compounds, and helium-nitrogen compounds in particular, as well as revealing a potential new single-bonded nitrogen structure that may be synthesizable in laboratory conditions. The analysis of bond lengths, charge density, electronic population and band structures, and phonon dispersion appears to be rigorous and sound, verifying the properties and stability of the predicted structures. I feel this paper presents enough interesting and novel content which falls within the scope of Nature Communications and the interests of its readers to recommend acceptance of this manuscript for publication*

REPLY: We are grateful to the reviewer for their work in reviewing our manuscript and the positive overall comments on our work.

REFeree#1 COMMENT 1. *It is mentioned that the weak interactions between the helium and nitrogen atoms in the $I4_1/a$ structure was a factor in considering whether the helium atoms could be removed and the nitrogen structure maintained. However, it also appears that the $C2/c$ structure also has weak He-N interactions. It may be worth noting why the $C2/c$ structure was not considered as well, whether it be due to the differences in the dimeric nitrogen in the $C2/c$ structure versus the polymeric, ring structure in the $I4_1/a$ structure, or some other reason.*

REPLY: The purpose of this work is to uncover the new polymeric form of pure solid nitrogen that may be important in technical applications such as those associated with high density energy materials. In contrast to the fully polymerized nitrogen framework with single N-N bonds that we find in the $I4_1/a$ structure, nitrogen atoms in the $C2/c$ structure form isolated diatomic molecular N_2 with triple $N\equiv N$ bonds. This phase is not, therefore, expected to not show high density energy storage, although it is a newly-predicted structure. This is the main reason why we do not carry out and describe further calculations on the stability of the $C2/c$ structure after removal of He atoms. For the same reasons we do not pursue these calculations for the $P\bar{1}$ structure either (since it contains both single N-N and double $N=N$ bonds). **We have added a comment at the bottom of page 5/top of page 6 to explain why we do not carry out this exploration in this paper.**

REFeree#1 COMMENT 2. *There are a number of spelling errors which should be addressed, for example, “choosed” instead of “chosen”, “metstably” instead of “metastably”, “axes” instead of “axes”.*

REPLY: We are grateful for the close reading of the manuscript by the referee and apologise that these typographic errors slipped through our first version. **We have corrected all spelling errors and tidied the text up in a number of places. We have also read the revised manuscript carefully to eliminate typographical and grammatical errors.**

Response to reviewer #2

REFeree#2 OVERVIEW. *This paper predicts several new potential high-pressure phases of nitrogen using density functional theory and the particle swarm optimization algorithm. A couple features make this work particularly novel.....This is an interesting contribution, the channel-containing nitrogen structure would be quite unusual, and while there is not experimental evidence for these phases yet, the results do suggest one might be able to make these compound by crystallizing a nitrogen helium mixture under pressure. So it may be publishable. However, there are some issues that need clarification before it would be suitable for publication.*

REPLY: We are grateful to the reviewer for their time reviewing our manuscript, and for the positive comments on our work. We believe we are able to provide clarification on all the points raised below.

REFeree#2 COMMENT 1. *The paper does not discuss the 1992 Nature report of an experimental $\text{He}(\text{N}_2)_{11}$ compound (DOI:10.1038/358046a0) or other nitrogen related inclusion compounds (DOI: 10.1063/1.3533957 & 10.1103/PhysRevB.64.184101). The $\text{He}(\text{N}_2)_{11}$ compound would be particularly interesting target for the authors to pursue, both in an attempt to clarify the structure of that experimentally reported structure and to substantiate their predictions for HeN_4 .*

REPLY: We are grateful for the referee for drawing our attention to these earlier papers. While there is a very wide range of nitrogen related inclusion compounds that we could refer to, we think that the work by Vos et al. (1992) is particularly interesting in relation to our predictions, and we have focused on this suggestion and incorporated reference to it towards the end of page 3. The experimental work by Vos et al. reports the synthesis at high pressure, and later work published in [PRB **83**, 134107 (2011)] reports structure information. This suggests that $\text{He}(\text{N}_2)_{11}$ has a hexagonal structure with 2 formula units (46 atoms) in a unit cell and orientationally-disordered N_2 molecules. We have built a hypothetical hexagonal structure for $\text{He}(\text{N}_2)_{11}$ based on the structure information provided in [PRB **83**, 134107 (2011)], and find it possesses a positive formation enthalpy of ~ 0.02 eV/atom as compared to a mixture of He and N_2 at 25 GPa. This could reflect inaccuracies in the crystal structure, but in addition the large number of atoms and the orientational disorder of N_2 molecules limit our ability to predict the crystal structure using current computational methods. Nevertheless, we expected that our newly predicted HeN_4 , with negative formation enthalpy, is synthesizable at high pressures. Our current results will certainly stimulate further experiment and theory to revisit such an important system at high pressures. **We have added discussion of the work of Vos et al. (1992) and that on other nitrogen-related inclusion compounds towards the end of page 3. We describe our attempts to determine the enthalpy of formation of the hexagonal structure for this stoichiometry at the bottom of page 8/top of page 9.**

REFeree#2 COMMENT 2. *The twin arguments that the helium is not strongly interacting with the nitrogen and that these phases would preferentially form over separate nitrogen and helium solids are perplexing. One would expect that a more densely packed nitrogen structure + pure He crystals would be more stable unless there is something special to the N-He bonding. So why does it form? Helium isn't exactly known for forming strong non-covalent interactions. It's hard to tell from the structure pictures--are any of these related to clathrate-type structures?*

REPLY: After removing the He atoms, the N framework of $t\text{-N}$ is still stable in terms of the phonon dispersion of new structure, meaning that it is dynamically stable structure. On decompression to

ambient pressure it can persist therefore as a metastable phase. We think, therefore, that the HeN₄ structure is kind of clathrate-type structure. Our enthalpy calculations suggest HeN₄ is more favorable than a mixture of He+N₂ at high pressures, indicating dense packing plays a critical role in helping to stabilize this structure. In these respects, it is akin to the previously-reported stable H₂O-He clathrate structure. **We have added comments regarding the clathrate nature of I4₁/a HeN₄ at the middle of page 9, to clarify this point.**

REFEREE#2 COMMENT 3. *I am not entirely clear what form of He/N are used in the formation enthalpies H(He) and H(N) when computing the stability. Are those crystalline? If so, which polymorph of N?*

REPLY: Here, crystalline phases of He and N were used for the formation enthalpy calculations. *hcp-He* [Physical Review 109, 328 (1958)] at 0, 25 and 50 GPa, *alpha-N₂* [Acta Cryst. B 30,929 (1974)] at 0 GPa and *epsilon-N₂* [J. Chem. Phys. 84, 2837 (1986); J. Chem. Phys. 93, 8968 (1990)] at 25 and 50 GPa were adopted. **In response to this comment, we have made clear the phases used in the calculations by adding the references to these structures in the caption of Figure 1.**

REFEREE#2 COMMENT 4. *Similarly, when the authors state that the low-pressure HeN₄ phase is stable all the way down to 8.5 GPa, is that stable relative to the epsilon molecular phase of nitrogen + crystalline He? It's not clear to me that the definition of stability used is one relative to the most stable phase at those same conditions. If it's not, then the important question is how much less stable is it than the thermodynamically preferred phase at those conditions? If it's too far above the stable molecular forms at those pressures, you won't have much hope of making even the metastable HeN₄ form.*

REPLY: In the decomposition calculations relative to solid He and N₂ (Figure 1d), the *hcp* phase of solid He was used in the pressure range of 0-130 GPa. For solid nitrogen, three different low-enthalpy structures were adopted at different pressures, the molecular *alpha* phase at 0 GPa, the molecular *epsilon* phase above 0 GPa, and the polymeric *cg* phase above 54 GPa. The phases of solid nitrogen used in this calculation were based on the phase diagram presented in Ref [J. Chem. Phys. 93, 8968 (1990)]. As can be seen from the phase diagram reproduced below, the *alpha* phase is the most stable structure at ambient pressure, and *epsilon* phase occupies a large stable pressure range below 60 GPa at 0 Kelvin, the temperature to which our calculations correspond. However, the *gamma* phase between the *alpha* and *epsilon* phases was not considered in our calculation due to its small stable pressure range. **The phases adopted are clarified in the revised caption of Figure 1.**

After careful evaluation, the formation enthalpy of HeN₄ relative to He and N is negative under pressure, indicating HeN₄ is more stable than pure solid He and N. The dense packing leads to a small PV (H=E+PV) part that plays an important role in stabilizing HeN₄ at high pressures. This also highlights that fact that pressure is a useful tool to help to synthesize new functional materials.

FIG. 1 taken from Ref [J. Chem. Phys. 93, 8968 (1990)]: Phase diagram of nitrogen

REFeree#2 COMMENT 5. How strongly is the He interacting with nitrogen lattice in these different phases? One could easily quantify the interaction energy by comparing single-point enthalpy differences of the different lattices with and without helium in the structure, analogously to how they studied the $I4_1/a$ phase without He.

REPLY: We thank the reviewer for this suggestion. **We have estimated the interaction energy of the three phases of HeN_4 with the equation $\Delta H_{\text{int}} = H_{\text{HeN}_4} - (H_{\text{N}} - H_{\text{He}})$. The results have been provided as Table S6 in the supplementary materials.** As seen below, the interaction between He and nitrogen framework in all three phases is relatively weak, in agreement with the conclusions suggested by ELF and charge transfer results.

Structure	P (GPa)	ΔH_{int} (eV/atom)
$C2/c$	25	-0.033
$P\bar{1}$	70	-0.012
$I4_1/a$	100	-0.07

REFeree#2 COMMENT 6. Minor point: I have not done Mulliken population calculations in the solid state, but they seem a potentially poor judge of charge transfer given their strong reliance on the nature of the underlying orbitals. This is very clear in Gaussian basis sets, where for example the amount of Mulliken charge transfer in the water dimer varies dramatically with the basis set, including even the direction of the charge transfer: Net Hartree-Fock charge transfer from hydrogen bond donor water to acceptor water molecule:

UNIVERSITY OF
CAMBRIDGE

Department of Earth
Sciences

cc-pVDZ -0.0430

cc-pVTZ -0.0292

cc-pVQZ -0.0245

aug-cc-pVDZ +0.0279

aug-cc-pVTZ +0.0038

aug-cc-pVQZ -0.0383

The solid state Mulliken analog requires localizing the planewave density onto local atomic like orbitals, and I am concerned that the results would depend very strongly on how one localizes the orbitals.

REPLY: The referee is correct to point out that calculations on charge transfer for a solid are indeed strongly related to basis set (e.g. Nature 457, 863–867 (2009)). Here, we simply aim to provide a qualitative analysis to arrive at an estimate of just how weak the charge transfer is between He and N in our HeN₄ structure. **We have emphasized the qualitative aspect of the calculation and limitations of the numbers in the revised caption of Table S3.**

Reviewers' Comments:

Reviewer #1 (Remarks to the Author):

The authors sufficiently addressed my questions and tidied up the manuscript. This paper presents enough interesting and novel content which falls within the scope of Nature Communications and the interests of its readers. I am glad to recommend acceptance of this manuscript for publication.

Reviewer #2 (Remarks to the Author):

The authors have addressed my earlier concerns in the revised version of the manuscript. They have clarified a number of the finer points, and the English grammar has been improved in many places.

Overall, they have identified a novel new phase of nitrogen and make good arguments for why it might be achievable experimentally, and I could imagine this work will stimulate new experiments. I think it is worthy of publication in Nature Communications.